# Exploring Visitors’ Visual Behavior Using Eye-Tracking: The Case of the “*Studiolo Del Duca*”

**DOI:** 10.3390/jimaging8010008

**Published:** 2022-01-09

**Authors:** Serena Mandolesi, Danilo Gambelli, Simona Naspetti, Raffaele Zanoli

**Affiliations:** 1Dipartimento di Scienze e Ingegneria della Materia, dell’Ambiente ed Urbanistica (SIMAU), Università Politecnica delle Marche, Via Brecce Bianche, 60131 Ancona, Italy; mandolesi@agrecon.univpm.it (S.M.); simona@agrecon.univpm.it (S.N.); 2Dipartimento di Scienze Agrarie, Alimentari e Ambientali (D3A), Università Politecnica delle Marche, Via Brecce Bianche, 60131 Ancona, Italy; danilo@agrecon.univpm.it

**Keywords:** art exploration, visual perception, visual patterns, eye-tracking, visitor experience, active vision

## Abstract

Although the understanding of cognitive disciplines has progressed, we know relatively little about how the human brain perceives art. Thanks to the growing interest in visual perception, eye-tracking technology has been increasingly used for studying the interaction between individuals and artworks. In this study, eye-tracking was used to provide insights into non-expert visitors’ visual behaviour as they move freely in the historical room of the “*Studiolo del Duca*” of the Ducal Palace in Urbino, Italy. Visitors looked for an average of almost two minutes. This study revealed which parts of the artefact captured visitors’ attention and also gives interesting information about the main patterns of fruition.

## 1. Introduction

The exploration of how the human brain perceives images has been studied since ancient times [1,2]. More recently, at the end of the 20th century, Zeki [3] provided a deeper explanation about the capacity of the human brain to form concepts and showed the connection between neural activity and visual stimuli viewing beautiful, neutral, and ugly images using fMRI.

Thanks to the evolution of visual analytic technologies, recent studies benefit from using eye-tracking as an essential tool in detecting previous insight into subjects’ visual behaviour. Eye-tracking, integrated with brain EEG and fMRI measures, has been increasingly used to detect and monitor subjects’ eye movements and visual pathways [4,5,6]. The study of visual behaviour using eye-tracking technology is based mainly on fixations and saccades. Fixations are brief moments in which our eyes are immobile, focusing on a particular part of a scene, painting, or photograph. Fixations, which usually range from 150 to 300 ms, relate to what we remember, how we learn, and people’s engagement in attention [5,7]. Saccades, which are rapid eye movements between two fixations, suggest shifts in attention [5]. Both fixations and saccades are excellent measures of visual attention and interest.

In the context of art, the use of eye-tracking is not new [1,8,9,10,11,12,13,14,15,16,17,18,19,20]. Quiroga et al. [13] studied participants’ differences in visual behaviour by comparing data collected viewing an original painting in the Tate Britain gallery and the digital image of the same artwork. Naspetti et al. [20] investigated the eye-movements of forty participants to define a new protocol for optimising an existing Augmented Reality (AR) application that allows the visualisation of digital content through a display.

In general, studies revealed that participants tend to focus on specific areas of the artefact (e.g., painting, photograph, etc.) according to the instructions provided and to the context [14,15,19,21]. Studies showed that images physical features, such as colour, luminance, or orientation, define visual saliency, which is a physical property of a visual item that makes it stand out from other items and immediately grab subjects’ attention [1,10,11,14,15,22]. Moreover, salient regions that are determined using specific algorithms [23,24] are inspected first by participants [14,25].

Although there are a relatively high number of eye-tracking studies about art, we know relatively little of how the human brain perceives art, especially in the context of museum experiences [13,26]. Moreover, there are few studies conducted inside museums and galleries [12,13,14,16,18,19,26]. With the development of eye-tracking on mobile devices, a new way for exploring an individual’s experiences in natural environments has been achieved [13,14,19,26]. According to Rainoldi et al. [26], mobile eye-tracking “*offers a non-invasive means to gather a window into what visitors see and what they do at a museum”.* Although most eye-tracking studies on art are performed in a laboratory with digital images and under controlled conditions, the use of a mobile eye-tracker offers some practical advantages. It allows analysing visitors’ behaviours in real-life conditions despite the individual looking “*at the original differently than a digital representation*” [13].

Visiting museums is a unique and complete experience. It allows for appreciating original artworks, capturing many details, such as the textural technique or the artist’s brushstrokes, or exploring the pieces from different viewpoints, also stimulating the curiosity of the visitors in historical buildings [13]. According to Balbi et al. [19], the context in which an artwork is viewed influences fruition strongly. Furthermore, the type of “*structure-container*” used to conserve artworks is relevant in attracting visitors [27]. In this framework, the visitor’s experience in the museum allows for a personal connection with the specific artwork [13,28].

In this study, mobile eye-tracking technology was used to provide insights into visitors’ visual behaviours in the historical room of the “*Studiolo del Duca*” of the Ducal Palace of Urbino, Italy. As they moved freely in the room, both the behaviours and eye movements of non-expert visitors were collected and analysed.

In Section 2, the participant sample, the equipment, the procedure, and the analysis are described. Section 3 presents the most relevant results, while in Section 4, the results are discussed.

## 2. Materials and Methods

### 2.1. Stimulus

The experiment took place inside the “*Studiolo del Duca*” of Federico da Montefeltro (*studiolo,* in what follows), which is part of the Ducal Palace of Urbino, one of the most important examples of Renaissance buildings in Italy (Figure 1).

The *studiolo* is famous for its iconographic importance and the inlays made with over forty different types of wood. This room can be observed as divided into two parts. The lower part of the walls displays inlaid images of both natural and abstract themes and objects: scientific instruments (e.g., the *mazzocchio*), musical instruments (e.g., organ), human figures of the three Theological virtues (Hope, Faith, and Charity), animals (e.g., squirrel, ermine, ostrich), the portrait and the armour of Duke Federico, and others [29]. In the upper part of the walls, there are portraits of famous men, known as “illustrious men”: political figures (e.g., Solomon, Seneca), poets (e.g., Homer, Virgil, Dante) and theologians (e.g., Duns Scotus) [29]. As investigated in Cheles [29] each element of the *studiolo*, as well as the relation between the portraits and some parts of the inlay panels, offers possible interpretation and encloses a precise meaning. More recently, the *studiolo* has been used as a case study in different research studies that used new technologies (e.g., X-reality technologies, augmented reality applications) to increase visitors engagement [27,30] in the context of Digital Cultural Heritage.

### 2.2. Equipment

Natural gaze behaviour data and eye position were recorded with the mobile eye-tracker Tobii Pro Glasses 2. This easy-to-use wearable eye tracker, now available in a new version, allowed us to observe and collect subjects’ visual behaviour from a first-person perspective. This eye-tracker weighed 45 g and was equipped with eye-cameras and a scene camera to track eye-movements and record where the participant is looking. The scene camera had an autofocus lens with a 90-degree diagonal field of view (16:9 format) and a resolution of 1920 × 1080 pixels at 25 frames per second. The accelerometer, included in the eye-tracker, allows for eliminating the impact of the head movements on eye-tracking data. Data were captured at 50–100 Hz. The quick calibration procedure was based only on one point. Each participant did their own personal calibration: they were asked to fixate the centre of a calibration card placed on a wall at a distance of 60 centimetres until the software confirmed the success of the calibration.

### 2.3. Sample and Data Collection

Prerequisites for participation in the experiment were: (a) no visual impairment (for example, wearing reading glasses, colour blindness), and (b) being a first-time museum visitor. A total of twenty-five participants were randomly picked from the museum visitors (fifteen males–aged between 24–77 and ten females–aged between 24–72). Participants’ level of education ranged from secondary school education to degree/postgraduate study. The majority of participants (64%) attended art classes at school.

Data were collected during two separate days during regular visiting hours in January 2020. Each participant gave written informed consent before participation. None of them received remuneration.

The experiment was conducted during the regular opening hours to reveal participant’s natural behaviour in an actual “museum situation”. Before the experiment, each participant calibrated the wearable eye-tracker following the calibration process previously explained. After this stage, each participant was instructed to look freely and move around naturally for as long as desired. No specific information about the *studiolo* was provided. In order to control the start of each participant, the viewing trial start was controlled and standardised, guiding participants at the entrance of the *studiolo*. The visit duration differed among participants (from 0.35 s to a maximum of 8 min and 9 s); however, excluding the extremes, the mean value of the visit was 1 min 44 s.

After the visit, all participants were asked to fill in questionnaires providing socio-demographic information, self-reported art expertise, interest in art, and the visitor experience assessment using a tablet. Based on a previous study on the assessment of visual art [31], participants expertise was assessed by asking them to answer specific questions about the frequency of art visits (“On average, how often do you visit art museums?” and “On average, how often do you visit art galleries?”), and about the time dedicated to visual art (“In the average week, how many hours do you dedicate to visual artistic activities?”, “In the average week, how many hours do you spend reading a publication that is related to visual art?” and “In the average week, how many hours do you spend each week looking at visual art?”). Results are shown in Table 1.

The other two constructs were measured, adapting reliable scales proposed by [32,33,34]. The first construct was aimed to assess participants experience (using a scale adapted from Pine et al. [32]), and the second construct aimed to measure individuals’ levels of interest in art (using a scale adapted from [33,34]). Each scale included twenty items (Table 2) rated from 1 (strongly disagree) to 7 (strongly agree) by using a 7-point Likert scale. The scale construct reliability was tested with Cronbach’s α (reliability for both scales confirmed with α = 0.82). Finally, socio-demographic questions concluded the survey.

### 2.4. Analyses

Two types of analysis were developed: an empirical analysis based on eye-tracking, aimed at collecting visual data for AOI from the direct experience of participants’ visit of the *studiolo*, and a computer-based analysis aimed to provide saliency maps of specific parts of the *studiolo* room in order to identify possible salient regions.

The empirical analysis was conducted using the following software: Imotions^®^ Attention Tool (vers. 8.1) and Stata (vers. 17). Data collected with the mobile eye-tracker were aggregated and mapped onto static images of the particular stimuli encountered during the recording and relevant metrics were measured by Imotions^®^ Attention Tool^®^. A synopsis of eye-tracking metrics relevant for analysing artworks is reported in Table 3 [10,19,20]. Heatmaps and gaze plots are data visualisations that allow for identifying relevant aspects of a subject’s visual behaviour. Gaze plots show the position and the order on the stimulus of the sequence of looking. Heatmaps show how looking is distributed over the stimulus revealing the focus of visual attention. In the heatmap, “*pure green*” is the lowest value indicating low attention, while “*pure red*” is the highest value indicating great attention [35]. The software provides different metrics for each AOI (area of interest). We focused on the Time spent-F, the TTFF-F, the Fixation count, and the participant-Ratio for this study. The Time spent-F provides the time spent in a specific AOI. The TTFF-F is the time of the first fixation inside a specific AOI since the participants entered the *studiolo*. A short time value of TTFF-F indicates that the participant’s fixation for that specific AOI started immediately as the participant entered the room, while a high time value shows that the fixation occurred at a later stage. The TTFF-F value is an important metric because it provides information about how certain aspects of a scene are prioritised. The Fixation count indicates the number of fixations within an AOI. The participant-Ratio indicates the number of participants that visited a specific AOI. The following paragraph describes heatmaps and AOIs of those areas of the *studiolo* that most attracted participants’ attention during the free visit. In this study, each AOI was centred and defined on a specific design element that is part of the *studiolo*.

Saliency maps, i.e., specific areas of a visual stimulus (saliency regions) that are noticeable from its neighbours, were defined using Matlab’s GBVS saliency model (Graph-Based Visual Saliency). Saliency maps are built based on software elaboration of the structural characteristics of images that are expected to attract most visitors’ attention [11,22].

## 3. Results

In this section, eye-tracking results are described focussing on the north and east walls of the *studiolo*, which were those that most attracted participants’ visual attention. Moreover, also for specific areas of the *studiolo*, a saliency-map analysis was conducted.

Figure 2 shows both the AOIs and the related metrics (on the left) and the heatmap of the north wall (on the right) for all participants. In this figure, AOIs are numbered from 1 to 10 according to the TTFF values (e.g., number 1 is assigned to that AOI with the lowest TTFF, while number 10 to that AOI with the highest TTFF). In the heatmap, red areas indicate relatively high attention and green areas relatively less attention. No colour indicates that the particular area is not observed at all. The heatmap shows which areas of the north wall were focused for the longest amount of time and which areas were missed by participants’ eyes.

Results concerning TTFF-F values indicate that participants focused on top of the wall where eight portraits of illustrious men are exposed. More specifically, participants’ attention was captured more by the four portraits of the illustrious men in the bottom row (Gregory the Great, St. Jerome, St. Ambrose, and St. Augustine), more visible with respect to the portraits placed above (see Figure 2). Among those, the portrait of St. Ambrose (AOI 1), located in a more central position, is firstly seen by visitors and accounts for the lower TTFF-F value (46.7 s). Then, the visitors’ eyes scanned the other portraits following this order: St. Jerome (AOI 2: TTFF-F = 54.7 s), St. Augustine (AOI 3: TTFF-F = 60.1 s), Moses the Jew (AOI 5: TTFF-F = 78.2 s), that portrait is placed in the east wall near the corner and very close to St. Augustine, and Gregory the Great (AOI 6: TTFF-F = 79.9 s) in the left side corner. It should be noted that the TTFF-F value of St. Ambrose’s portrait was significantly lower than those of Gregory the Great (*p* < 0.01) and of Moses the Jew (*p* < 0.05) paintings. Although the portrait of St. Ambrose was the first seen by visitors, St. Jerome’s portrait (which is a copy of the original conserved at the Musée du Louvre, Paris) accounted for the highest number of fixations (143). The high number of fixations is probably due to the high salience of this portrait, which is exceptionally bright and uncoloured compared to the others. Data for each AOI of the north wall are ported in Table 4.

The saliency map in Figure 3 shows how the most salient area refers to the portrait of St. Jerome. This portrait copy, placed next to St. Ambrose’s painting in a very central part of the north wall, actually attracted participants’ eyes. However, in terms of the number of fixations, there were significant differences (respectively 70 and 66 fixations, *p* < 0.05) with the painting of Gregory the Great and Moses the Jew.

According to the aggregated visual pattern based on the TTFF values (indicated in Figure 2a with the sequence of numbers from 1 to 10), participants, after observing some of the most central portraits of illustrious men (see AOIs 1, 2 and 3), moved their attention to the lower central part of the wall, where there are several inlaid elements (see AOI 4). More specifically, the heatmap shows that some objects placed in the cabinet highly attracted the visitors’ attention: a candle, an hourglass, and some books (AOI 4 Cabinet 2: TTFF-F = 77.4 s, 129 fixations). However, in terms of TTFF-F, there was only one significant difference between this cabinet and the AOI of Duke Federico (*p* < 0.01), which is the last seen by the visitors (AOI 10: TTFF-F = 109 s) and characterised by the smaller value of the participant-Ratio (only nine participants). The number of fixations for the AOI Cabinet 2 was significantly higher than that of all other elements placed in front of the entrance of the *studiolo* (i.e., Cabinet 1, Cabinet 3, and Duke Federico, with *p* < 0.05).

The participants’ gaze pattern shows they continued to turn to the exploration of the north wall returning up to the portraits of illustrious men placed near the extremes (see AOIs 5 and 6) and then jumping down and focusing on the rest of the wooden panel (see AOIs 7, 8, 9, and 10 in Figure 2a). According to this sequence, participants’ eyes focused on the Theological virtue of Hope (AOI 7: TTFF-F = 93.9 s), then on Cabinet 1 (AOI 8: TTFF-F = 90 s), Cabinet 3 (AOI 9: TTFF-F = 102.4 s), and finally on the figure of Duke Federico (AOI 10: TTFF-F = 109 s). The human figure of the Hope virtue was characterised by the longest Time spent-F (0.7 s), see Figure 4. No statistical difference was found between this virtue and the other AOIs of the same wall area in terms of time spent. By contrast, in terms of fixation, the virtue accounted for the highest number of fixations (131) compared to the AOIs of Cabinet 1 and Cabinet 3 (with *p* < 0.05) (AOIs: Cabinet 1: 61 fixations, AOIs: Cabinet 3: 42 fixations).

The painting of Moses the Jew, placed on the east wall, attracted participants’ attention. The interest in this painting is also confirmed by the heatmap presented in Figure 5b. Despite this, there were no statistical differences between this painting and the other portraits of illustrious men of the east wall (i.e., Cicero, Seneca, Homer, Virgil, Solomon, St. Thomas Aquinas, Duns Scotus), this painting accounted for the highest number of fixations (199) and the higher Time spent (1.1 s). AOI metrics of the east wall (upper part) are presented in Table 5. However, the saliency map in Figure 6 indicates that the light-blue background of this portrait of an illustrious man is particularly salient.

Focusing on the lower part of the east wall, Figure 7 shows which motifs of the inlaid panel captured participants’ gaze. The first element seen by participants and the one reporting the highest number of fixations was the squirrel (AOI 1: TTFF-F = 46.2 s, 196 fixations). More specifically, the number of fixations for this element was significantly higher compared to those of the basket of fruit (AOI 4: 102 fixations), the cabinet (AOI 6: 29 fixations), or the abstract patterns (AOI 3: 64 fixations), with *p* < 0.001 and, compared to the Duke’s armour (AOI 5: 90 fixations), with *p* < 0.05. Moreover, the TTFF-F value of the squirrel was significantly lower than those for the landscape (*p* < 0.05), the cabined (*p* < 0.001), the abstract patterns (*p* < 0.01), the armour (*p* < 0.05), and for the miniature *studiolo* (*p* < 0.01). The other element which attracted participants’ eyes was the landscape depicted in the centre of the civic space (AOI 2: TTFF-F = 59 s, 136 fixations). Despite the squirrel being the first element seen by participants, participants spent a longer time viewing the landscape area. However, no statistical difference was found between the landscape and the other AOIs in time spent. According to the participants’ gaze pattern, defined by the sequence of the numbers from 1 to 7 (Figure 7a), the AOIs Cabinet, Armour, and Miniature *studiolo* were lastly explored. Particularly, the lower area, where there is the open door of one cabinet, seems to be less attractive for participants (AOI 6: 29 fixations), accounting for the smaller number of participants that looked into this AOI (only 6). Data for each AOI are ported in Table 5.

According to eye-tracking data, gaze data show that participants were more attracted by the north and east walls, spending less time on the other two walls. However, it was not possible to define a common viewing pattern among participants. Despite all participants entering the *studiolo* from the same entrance, their eye movements were markedly different. For example, one started looking at the woodwork from the right side where there is the miniature of the *studiolo*, and then focused on the inlays of the north wall. Another started looking on the left side and then moved to the portraits of the illustrious men placed above the inlaid panel of the north wall, etc. However, the analysis allowed us to find one salient difference between the north and the east wall pattern: when looking at the north wall, all participants—except participant 1, who focused more on the east wall—were captured by the upper part (where the portraits of the illustrious men are located), while considering the east wall they focused more on the lower part where there are inlay decors. In other words, the portraits of the illustrious men represented the main “attractors” for participants’ eyes in the north wall, while this was not true for the east wall.

In general, the viewing pattern of participants varied across the walls. Most participants tended to express a different visual behaviour viewing the diverse parts of the *studiolo*. Both Figure 8 and Figure 9 show gaze plots of three distinct participants for the north and east walls. For example, participant 3, when observing the north wall, ignored the inlaid panel and focused more on the upper part where there are portraits of the illustrious men, while on the east wall, their eyes explored both the inlaid panel and some portraits of illustrious men located in the left corner. Participant 7 explored the north wall almost entirely, while the east side was observed only on the lower part where there are inlay decors. Meanwhile, participant 18 focused mostly on the portraits of the illustrious men looking at the north and the east wall.

## 4. Discussion and Conclusions

In this study, eye-tracking technology was used to identify which parts of the *studiolo* attract the most attention and explore participants’ viewing pathways. Results presented in this study may contribute to visual behaviour analysis on art in the original context, as the majority of eye-tracking studies have been conducted in laboratories using digital copies. Furthermore, this is the first eye-tracking study conducted at the “*Studiolo del Duca*” of Urbino (Italy), providing a new “exploration” of this famous ambient.

Results indicated that the duration of the visitors’ experience was relatively short, less than two minutes. This aspect confirms that the natural observation of artwork inside a museum is very short: in this case, visitors have a fairly short look around the room before going forward [36,37].

The eye-tracking data revealed that the elements which received the highest number of fixations were: (a) human-contents (e.g., paintings of the illustrious men, the Hope virtue, Duke Federico); (b) more dynamic and contrasted elements (e.g., the squirrel of the civic space). This result supports the idea that in lack of information, the human figures drive participants’ attention, indicating their role as “attractors” [15]. More specifically, the lower TTFF-F values for the human figures confirmed the more substantial power of the human content to dominate participants’ visual behaviour, compared to the other unanimated motifs depicted in the *studiolo*. In general, when focusing on human figures, results confirmed that the faces are the stronger attractors, given that it is the first part of the body that drew visual attention [8,15,38]. For example, looking at the heatmap of the north wall (Figure 2b), the most relevant red area is the one reported over the face of the Hope virtue, followed by the other two faces (see red areas over St. Jerome and St. Ambrose). Moreover, according to our results, in the absence of human content—as in the east wall—other more dynamic elements (e.g., the squirrel) captured participants’ eyes more than the landscape or the basket of fruit, which are intrinsically more “static” [15], see Figure 7.

Those results can also be correlated with another relevant element: the vanishing point [38]. As reported in Crucq [38], visitors viewing an artwork seem to be attracted by the vanishing point, especially if “*it coincides with the central vertical axis of the painting, but is even stronger when the vanishing point also coincides with a major visual feature such as an object or figure*”. It is not a case that the Hope virtue accounted for the highest number of fixations (131) for the north inlay panel. According to Raggio [39], the figure of the Hope virtue represents the vanishing point for the north inlay panel: “*The figure of Hope is the vertical axis and the vanishing point of a perspectival scheme that includes the benches as well as the two pairs of cupboards*”.

For the east wall, something changed. In the east inlay panel, “*the orthogonals are marked in the foreground of the central panel, and the vanishing point is on the horizon of the distant landscape*” [39], the squirrel is the element that attracted the most views of participants, with 196 fixations. This confirms that when the vanishing point is in the central axis but does not coincide with a figure, fixations are higher “*on the figures and on their faces*” [38], (see the red area over the squirrel of the heatmap in Figure 7b).

According to previous studies [12,19], results confirmed that there is not a common or typical viewing pattern since participants’ visual pathways were significantly different. This wide variety may be associated with the *studiolo* complex 3D ambient and with the absence of a clear task of visit for participants.

Exploring an ambient like the *studiolo* is necessarily different from the observation of a single artwork like a painting. The irregular plan, the presence of darker sides (on the west and south wall), and the position of the spotlights—which cause reflections—influences the visibility of some particulars, increasing the observation of those elements with a more favourable position. Participants mostly ignored elements of the west and south sides.

According to literature, without a specific interest or task at hand, visitors’ attention is stimulus-driven, attracted by low-level visual features such as an intense colour or a high contrast [10,11,14,15]. Some portraits of the illustrious men attracted first the visual attention of participants with respect to the lower part of the *studiolo*, where the wooden panels are substantially “homogeneous” in colour. The copy of St. Jerome accounted for the highest number of fixations in the north wall. In this case, the dominance of the white colour probably contributed to the increased attention towards this portrait. Similar considerations can be taken for the painting of Moses the Jew of the east wall. Surprisingly, despite the saliency map of the east wall (Figure 6), which indicated the light-blue background as the most salient part, visitors’ attention towards this painting, indicated by the relatively high number of fixations and by the long time spent on this painting, can be mainly associated with the Arabic inscriptions present in the book. Visitors spent a longer time on that painting because they tried to read the inscriptions. The high attention towards those inscriptions is highlighted in the heatmap with the red colour (see Figure 5a and Table 5). This result is in line with previous literature suggesting that both texts [40] and complicated and unclear images [6] attract more attention in a free viewing task requiring higher cognitive processing and mental workload.

The study has some limitations that should be acknowledged. First, participants were asked to look at the *studiolo* freely without specific tasks. Visitors’ behaviour could be investigated by testing visitors after having provided similar information about the artefact. This would allow exploring the role of top-down and bottom-up attentional processes. Another exploratory aspect, which has not yet been fully developed, is the study of correlations between the duration of the visiting time and art appreciation.

Nevertheless, this study shows that eye-tracking is an effective technology for analysing how visitors “consume” artworks in museums. In the last years, museums and art galleries have increased their efforts investigating new ways to improve visitors’ engagement and increase the time spent in front of an artwork [36]. The precious information gained with eye-tracking techniques can be relevant in the process of reinventing the customer experience (e.g., increase time spent, provide personalised experience) because it offers the possibility to explore the essential factors of a museum visit. Further, gaze data may also suggest how to better allocate artefacts and paintings or increase their visibility when possible. For example, this analysis suggested how a better illumination could improve the quality of the artefact’s exposure to visitors. It is known that lighting is a critical component in a museum environment because it allows creating the right atmosphere without damaging artefacts [41]. In the *studiolo* room, there are several spotlights installed in the lower part of each wall. This type of illumination causes reflections which are particularly annoying for visitors, especially when the light source is present in a dark background [42]. Probably, those reflections negatively impacted the participant’s visual comfort, gaze, and duration [41,43].

Finally, further studies could consider a specific analysis of visual experience for subsamples of visitors, differentiated, e.g., in terms of frequency of museum visits, cultural interests, etc.

## Figures and Tables

**Figure 1 jimaging-08-00008-f001:**
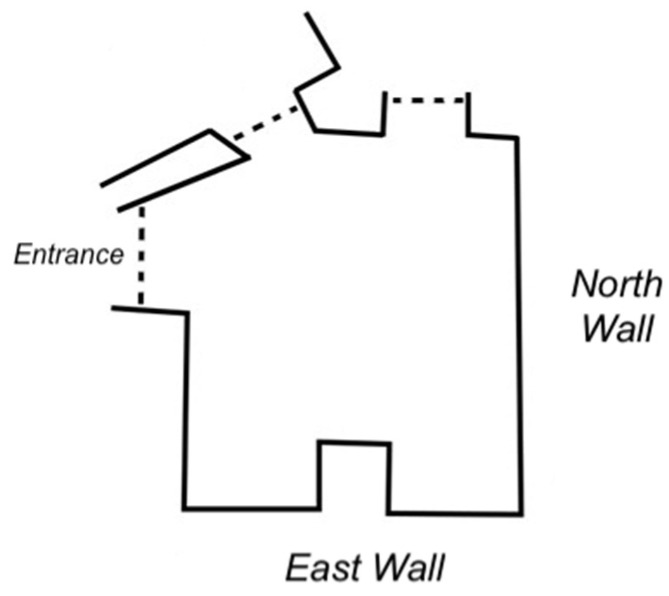
Plan of the “*Studiolo del Duca”,* Ducal Palace of Urbino.

**Figure 2 jimaging-08-00008-f002:**
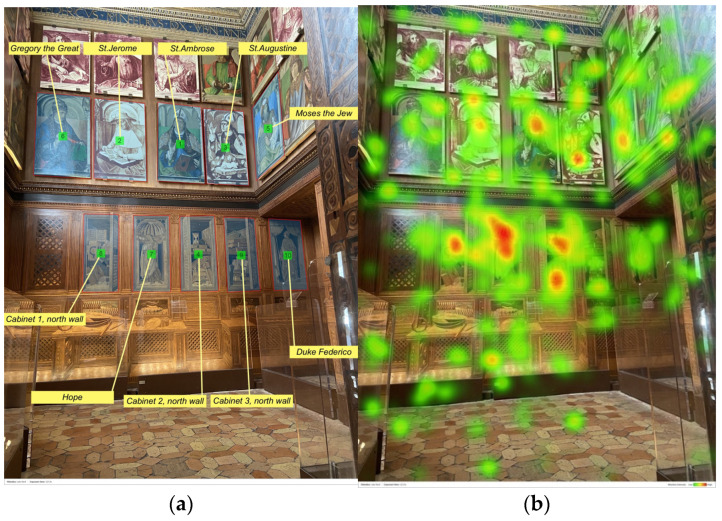
North wall: (**a**) AOIs; (**b**) heatmap.

**Figure 3 jimaging-08-00008-f003:**
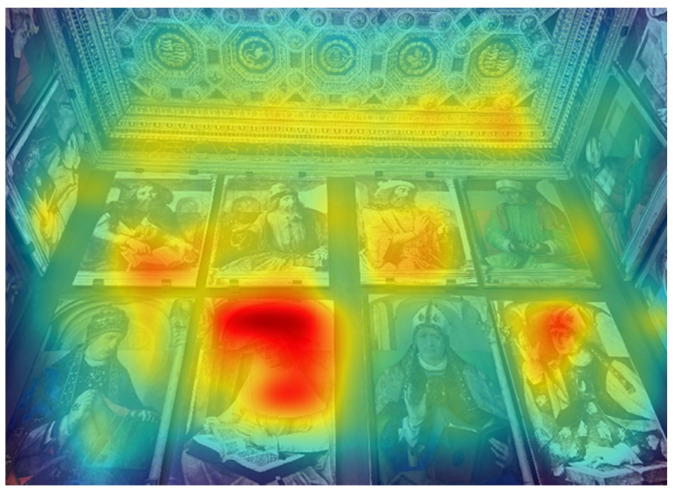
North wall: saliency map (portraits of the illustrious men).

**Figure 4 jimaging-08-00008-f004:**
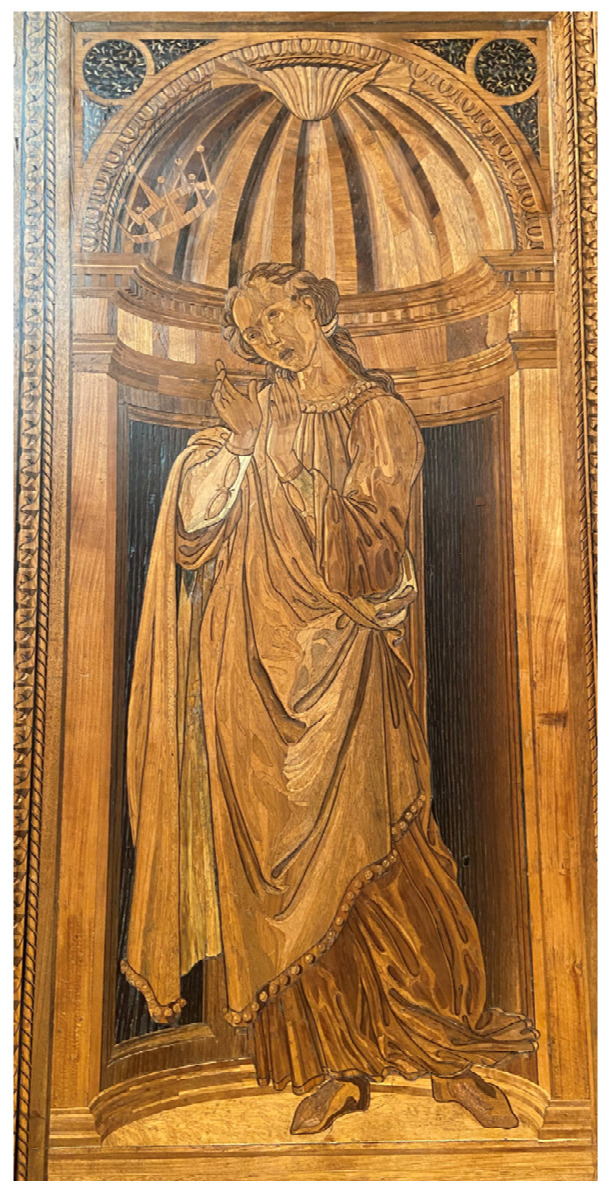
North wall detail: the Hope virtue.

**Figure 5 jimaging-08-00008-f005:**
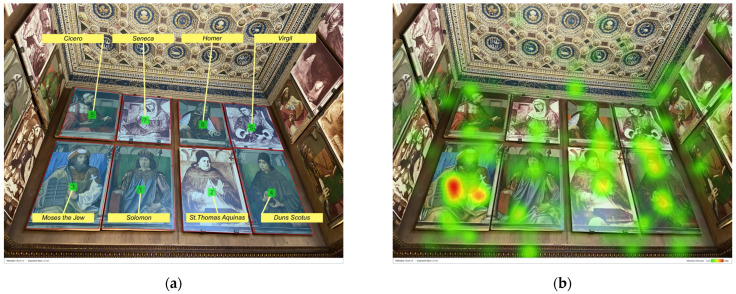
East wall (upper part): (**a**) AOIs; (**b**) heatmap.

**Figure 6 jimaging-08-00008-f006:**
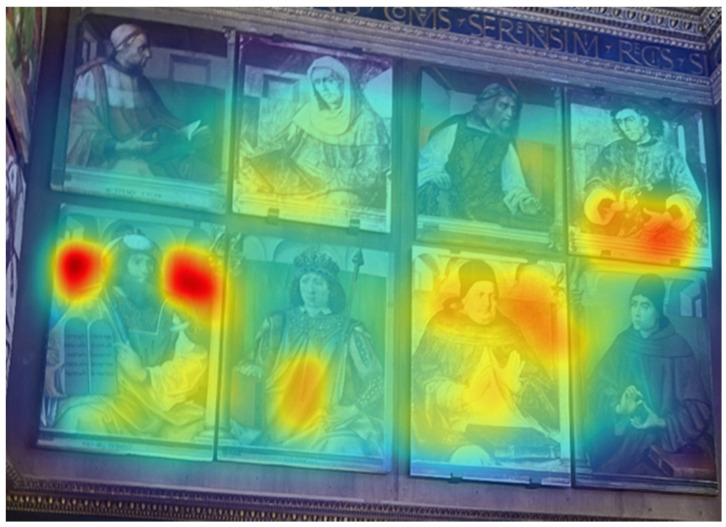
East wall (upper part): saliency map (portraits of the illustrious men).

**Figure 7 jimaging-08-00008-f007:**
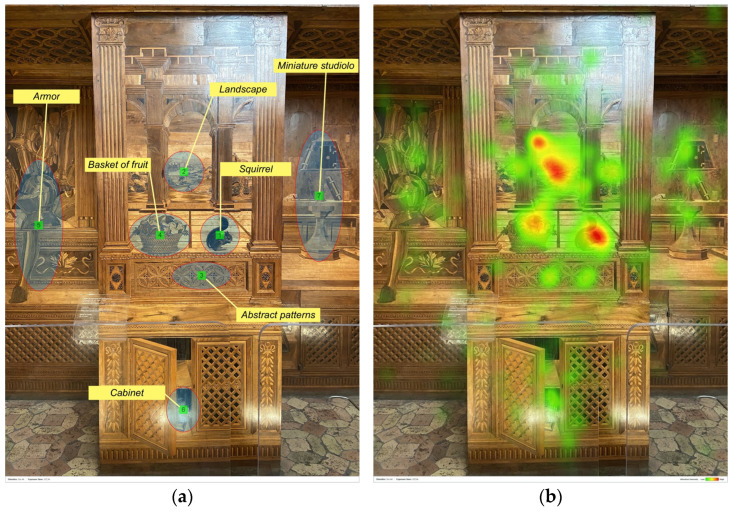
East wall (lower part): (**a**) AOIs; (**b**) heatmap.

**Figure 8 jimaging-08-00008-f008:**
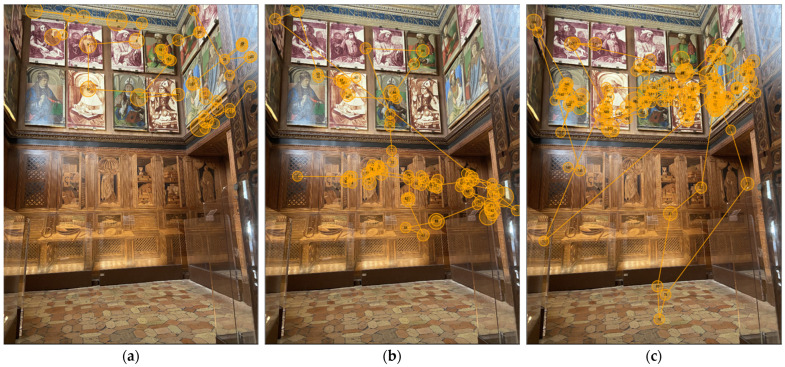
North wall: gaze plots for (**a**) participant 3; (**b**) participant 7; (**c**) participant 18.

**Figure 9 jimaging-08-00008-f009:**
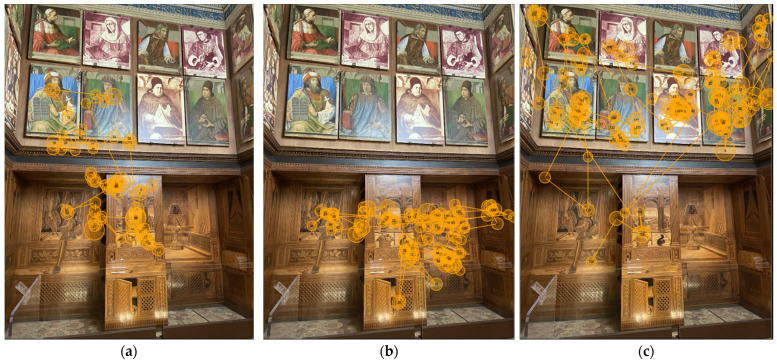
East wall: gaze plots for (**a**) participant 3; (**b**) participant 7; (**c**) participant 18.

**Table 1 jimaging-08-00008-t001:** Participants’ experience in museums and galleries visiting (relative frequencies).

	Almost Never	Once a Year	Every Six Months	Every Two Months	Every Month	Every Week	
On average, how often do you visit art museums?	4%	16%	36%	28%	12%	4%	
On average, how often do you visit art galleries?	8%	28%	24%	24%	8%	4%	
	0 h	1 h	2 h	3 h	4 h	5 h	6 h
In the average week, how many hours do you dedicate to visual artistic activities?	16%	44%	8%	12%	4%	4%	12%
In the average week, how many hours do you spend reading a publication that is related to visual art?	32%	36%	12%	4%	0%	4%	12%
In the average week, how many hours do you spend each week looking at visual art?	16%	52%	16%	4%	0%	4%	8%

**Table 2 jimaging-08-00008-t002:** Interest-in-art scale and experience scale (R means “reversed scale”).

	Interest in Art-Scale Items		Experience-Scale Items
1	Whenever I see a poster related art, I check it out	1	This experience stimulated my curiosity
2	I read culture and art pages of newspapers	2	This experience increased my knowledge
3	I do not like talking about art with my friends (R)	3	This experience enhanced my philosophy of living
4	I am interested in a branch of art unprofessionally	4	I like sharing this experience with my family and friends
5	I am not interested in painting exhibitions (R)	5	I like interacting with others in the museum
6	I could stare a long time at a beautiful painting	6	This experience relaxed me physically
7	I have a high appreciation for great architecture	7	This experience stimulated me emotionally
8	I think that individuals who deal with art are more creative	8	I had fun
9	I do not like reading book promotions of newspaper supplements	9	This experience was unusual
10	We talk and make discussions about art events in my family	10	I felt like someone else while in the museum
11	I believe that I should spare some money for artistic activities	11	I imagined living in a different time and place
12	When I see a beautiful photograph, I want to obtain information about it	12	At the museum I avoided interactions with others
13	I do not like following artistic events on the Internet (R)	13	I escaped from reality
14	I think that art is necessary for individual development	14	I wanted to get away from crowds of people
15	I watch carefully when there is news on TV about art	15	I wanted to get away from a stressful social environment
16	I do not like buying books about art (R)	16	I felt a sense of harmony with my surroundings
17	I’m passionate about art	17	This museum has a pleasing physical environment
18	I like decoring the walls of my room with nice artwork	18	Exhibitions are pleasant
19	I like doing research about artists and their works on the internet	19	I appreciated the different artefacts
20	I like going to exhibitions with friends and experts of art	20	I felt a pleasant interior ambience

**Table 3 jimaging-08-00008-t003:** Eye-tracking metrics and tools (source: www.imotions.com, accessed on 6 December 2021).

	Description
Gaze points	Constitute the basic unit of measure. Show what eyes are looking at. In figures identified by numbers.
Gaze plots	Show the position and the order on the stimulus of the sequence of looking.
Fixations	Period in which eyes are locked toward a specific object. Typically is 100–300 ms.
Saccades	Eye movements between fixations.
Heatmaps	Aggregation of gaze points and fixations showing the general distribution of visual attention. Typically displayed as a colour gradient overlay on the stimulus.
AOIs	Areas of interest are user-defined subregions of a displayed stimulus used to extract metrics.
TTFF-F	Time to first fixation indicates the amount of time that it takes a respondent (or all respondents on average) to look at a specific AOI defined from a visual stimulus as the respondent entered the room.
Time spent-F	Quantifies the amount of time that respondents have spent looking at a particular AOI.
Fixation count	Indicates the number of fixations within a specific AOI.
Ratio	Provides information about how many of your respondents actually guided their gaze towards a specific AOI.

**Table 4 jimaging-08-00008-t004:** AOI metrics for the north wall (see Figure 2).

North Wall										
AOI	1	2	3	4	5	6	7	8	9	10
	St. Ambrose	St. Jerome	St. Augustine	Cabinet 2	Moses the Jew	Gregory the Great	Hope	Cabinet 1	Cabinet 3	Duke Federico
TTFF (s)	46.7	54.7	60.1	77.4	78.2	79.9	93.9	90	102.4	109
Time Spent (s)	0.4	0.5	0.3	0.5	0.3	0.3	0.7	0.3	0.2	0.2
Ratio	23/25	20/25	16/25	17/25	12/25	15/25	15/25	10/25	10/25	9/25
Fixations	124	143	82	129	66	70	131	61	42	71

**Table 5 jimaging-08-00008-t005:** AOI metrics for the east wall (both lower and upper parts, see Figure 5 and Figure 7).

**East wall (lower part)**								
**AOI**	**1**	**2**	**3**	**4**	**5**	**6**	**7**	
	**Squirrel**	**Landscape**	**Abstract Patterns**	**Basket of Fruit**	**Armour**	**Cabinet**	**Miniature Studiolo**	
TTFF (s)	46.2	59	69.6	75.1	88.4	91.8	101.3	
Time Spent (s)	0.6	0.7	0.3	0.5	0.2	0.1	0.4	
Ratio	20/25	15/25	11/25	16/25	13/25	6/25	11/25	
Fixations	196	136	64	102	90	29	110	
**East wall (upper part)**								
**AOI**	**1**	**2**	**3**	**4**	**5**	**6**	**7**	**8**
	**Solomon**	**St. Thomas Aquinas**	**Moses the Jew**	**Duns Scotus**	**Cicero**	**Virgil**	**Seneca**	**Homer**
TTFF (s)	65.3	72.5	72.9	88.4	92.1	100.1	103.7	109.3
Time Spent (s)	0.5	0.5	1.1	0.6	0.2	0.2	0.1	0.2
Ratio	18/25	18/25	15/25	12/25	11/25	10/25	8/25	7/25
Fixations	160	154	199	144	38	38	20	32

## Data Availability

Data presented in this study are available on request from the corresponding author.

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
