# Peer review of "Exploring Visitors’ Visual Behavior Using Eye-Tracking: The Case of the “Studiolo Del Duca”"

_2313-433X, 2022, doi:10.3390/jimaging8010008_

Round 1
Reviewer 1 Report
This article presents the study of the visual behavior of non-expert visitors using mobile eye-tracking technology in the historical room of the “Studiolo del Duca” of the Ducal Palace in Urbino, Italy.
Since just a few eye-tracking studies have been conducted inside museums and galleries the study presented in the paper is of high interest. The study employs mobile eye tracking technology to explore the natural gaze behaviour of twenty-five randomly picked.
Overall, the paper is well written and provides a good content to the body of knowledge. The paper can be published after minor revision of the following:
- There is a missing full stop in line 72.
- There are missing references between the lines 155 and 160.
- The text within Figures 2a, 4a and 6a is too small and difficult to read.
- In the discussion section it should be mentioned how mobile eye tracking studies like the one just presented can extract important information to inform museums and art galleries about strategies to better allocate artifacts and paintings to increase their visibility, improve visitors experience, and increase time expend.

Author Response
We appreciate the reviewer’s kind assessment of our paper. We considered all points and modified the text as suggested.
- Line 72 was corrected.
- References were added for lines 155-160.
- Text for each Figure was adjusted to improve readability.
- Improvements required were added to the discussion and conclusion
Thank you.
Reviewer 2 Report
The introduction provides appropriate background and includes relevant references. References are rich and updated. They stimulate curiosity to deepen the topics presented in the paper.
In my opinion, the fact that the experiment took place inside the “Studiolo del Duca” of Federico da Montefeltro, which is part of the Ducal Palace of Urbino, adds to the whole research a special interest for the reader. The research design seems adequate to the publication in this Journal.
Moreover, methods presented are appropriated to a scientific study.
Finally results and conclusions are clearly presented and seem of interest for the final reader.
English language is appropriate and used with fluency. Nevertheless, my advice is to read the whole text to find out any possible typos.
The paper is original, well written, significant and clear. It is of interest to the readers and appropriate for the Journal.
Author Response
Thank you!
Reviewer 3 Report
- There are a few typos that need to be corrected.
- Where were the scales used? Statistical analyzes seem to have been done only for eye-tracker data. For example, is there a difference between expert visitors and non-experts?
- What is recommended as a result? Could a new layout or similar improvement be possible to make it easier for non-expert visitors to browse the artworks?
Author Response
R. There are a few typos that need to be corrected.
Q. We checked the manuscript and corrected the typos
R. Where were the scales used? Statistical analyzes seem to have been done only for eye-tracker data. For example, is there a difference between expert visitors and non-experts?
A.
We apologize if we did not clarify the use of scales in the paper.
Information about the experience and interest in the art of the participants were used to better define the visitors’ sample. However specific analyses of the results for different subsamples were beyond the scope of our analysis but may be considered for further studies. This is now included in the discussion and conclusion section
R. What is recommended as a result? Could a new layout or similar improvement be possible to make it easier for non-expert visitors to browse the artworks?
R.
In the discussion, we have now clarified how eye-tracking data may increase visitors’ experience by providing relevant information for museums and galleries.
In the case of the “studiolo” (an authentic room in an ancient building), changes in the layout are not possible. However, results suggest patterns of fruitions for the potential visitors, not available before this study, and improvements for the room's illumination.
Thank you for the valuable comments.